# High-throughput preparation of radioprotective polymers via Hantzsch's reaction for in vivo X-ray damage determination

Guoqiang Liu[1], Yuan Zeng[1], Tong Lv[2], Tengfei Mao[1,3], Yen Wei[1], Shunji Jia[2], Yanzi Gou[3] & Lei Tao [1✉]

Radioprotectors for acute injuries caused by large doses of ionizing radiation are vital to national security, public health and future development of humankind. Here, we develop a strategy to explore safe and efficient radioprotectors by combining Hantzsch's reaction, high-throughput methods and polymer chemistry. A water-soluble polymer with low-cytotoxicity and an excellent anti-radiation capability has been achieved. In in vivo experiments, this polymer is even better than amifostine, which is the only approved radioprotector for clinical applications, in effectively protecting zebrafish embryos from fatally large doses of ionizing radiation (80 Gy X-ray). A mechanistic study also reveals that the radioprotective ability of this polymer originates from its ability to efficiently prevent DNA damage due to high doses of radiation. This is an initial attempt to explore polymer radioprotectors via a multi-component reaction. It allows exploiting functional polymers and provides the underlying insights to guide the design of radioprotective polymers.

[1] The Key Laboratory of Bioorganic Phosphorus Chemistry and Chemical Biology (Ministry of Education), Department of Chemistry, Tsinghua University, Beijing 100084, China. [2] State Key Laboratory of Membrane Biology, Tsinghua-Peking Center for Life Sciences, School of Life Sciences, Tsinghua University, Beijing 100084, China. [3] Science and Technology on Advanced Ceramic Fibers and Composites Laboratory, National University of Defense Technology, Changsha 410073, China. ✉email: leitao@mail.tsinghua.edu.cn

Owing to the wide use of nuclear technology in modern society, accidental nuclear leakages and potential terrorist attacks considerably increase the risk of exposure to high doses of ionizing radiation[1]. Meanwhile, in a recent NASA twins study, it was confirmed that one of two brothers who stayed in outer space for just 340 days had 8–9% of his DNA permanently mutated because of the strong ionizing radiation[2]. This indicated that, in terms of radioprotection, humans are not ready for space travels that may require years or even decades. Radioprotectors have been studied for more than 60 years, as the applications of ionizing radiation in the energy, medicine and military fields[3]. Considerable results have been achieved in this area, as demonstrated by amifostine, which is a phosphorothioate that was explored by the Anti-radiation Drug Development Program of the U.S. Army. This compound has been approved as the only radioprotector for narrow clinical indications associated with radiotherapy[3–5]. However, amifostine is rapidly excreted by the human body (only ~5% left in the plasma 1 h after its administration) and has serious side effects (e.g. hypotension, fever, nausea and vomiting) even at low doses[6,7]. This considerably limits its application to counteract injuries caused by high doses of ionizing radiation. Thus, safe and effective radioprotectors to treat or prevent acute injuries induced by high doses of radiation are urgently needed for the future of humankind.

Including small-molecule drugs into polymeric structures is a straightforward method to solve some problems associated with the use of these small molecules (e.g., quick elimination from the body, poor water solubility, instability and toxicity). Nevertheless, there are only very few studies on polymeric radioprotectors for the following reasons. (1) Some functional groups, such as thiols, nitroxides and bis-benzimidazole, have been identified as promising radioprotective groups[3,5,8]. Preparing monomers containing these groups is a direct method to exploit radioprotective polymers. However, including these groups and their derivatives into monomeric structures typically requires laborious multi-step reactions. This considerably increases the difficulty and cost of synthesis. (2) In addition to the anti-radiation ability, many other factors (e.g., safety and bioavailability) should be comprehensively evaluated when exploring a radioprotector. Thus, an applicable radioprotector is normally obtained by screening a library containing many candidates. For example, ~4400 aminothiols and their phosphothiolates have been prepared and tested to produce amifostine[3]. This is difficult to duplicate for the development of polymeric radioprotectors owing to the lack of simple methods to quickly prepare monomer/polymer libraries.

Recently, many multi-component reactions (MCRs) have been used to prepare polymers. These MCRs include Passerini, Ugi, Biginelli, Hantzsch, Kabachnik–Fields and Mannich reactions[9–24]. We believe that MCRs can help develop polymeric radioprotectors, because some MCRs can generate products with a considerable anti-radiation ability (e.g., the Hantzsch, Biginelli and Kabachnik–Fields reactions)[25–27]. Thus, polymers prepared by these MCRs may be potential radioprotectors. Meanwhile, our previous studies confirmed that MCRs are powerful and can easily prepare monomer/polymer libraries in a high-throughput (HTP) manner[28]. This approach may overcome the restrictions in developing safe and effective polymeric radioprotectors.

Here we report a polymer radioprotector prepared by the combination of Hantzsch's reaction, HTP technology and polymer chemistry, and its application in protecting cells and zebrafish embryos from high doses of ionizing radiation (80 Gy X-ray) (Fig. 1).

Hantzsch's reaction includes four common components (i.e., aldehyde, 1,3-diketone, β-ketoester and NH$_4$OAc) to effectively produce 1,4-dihydropyridines (1,4-DHPs). This reaction was first reported by Arthur R. Hantzsch in 1881[29] and has been broadly studied in the fields of pharmaceutical chemistry and organic chemistry, because 1,4-DHPs are candidate drugs for treating cardiovascular diseases[30,31]. Recently, this reaction has been used in polymer chemistry[23,32]; however, monomer libraries prepared via Hantzsch's reaction are not very common.

In this study, a library of monomers has been efficiently synthesized via Hantzsch's reaction in an HTP manner with high yields. These monomers have been used to construct a library of water-soluble polymers via HTP copolymerization with a water-soluble monomer. These polymers have been then screened by HTP measurements to achieve a biocompatible polymer with the best anti-radiation capability. In cellular and in vivo experiments, this selected polymer has efficiently protected cells and zebrafish embryos from lethal doses of ionizing radiation (80 Gy X-ray). Hence, its protective effect is superior to that of amifostine. This suggests the utility of MCRs and the HTP strategy in exploiting biofriendly polymer radioprotectors for possible practical applications.

## Results and discussion

**HTP preparation of the monomer library via Hantzsch's reaction and a related polymer library.** A commercially available monomer, 2-(acetoacetoxy)ethyl methacrylate, was converted into monomers containing different 1,4-DHP moieties via Hantzsch's reaction in an HTP manner. Using different combinations of nine aldehydes (A(X)) and five 1,3-cyclohexanedione derivatives (B(Y)), 45 (9 × 5) Hantzsch monomers (M(X)(Y)) were simultaneously created (Fig. 2a and Supplementary Figs. 1–4).

The target 45 monomers were easily obtained with high yields (88–98%) after simple precipitation. As a typical example, the $^1$H nuclear magnetic resonance (NMR) spectra of the M(X)(1) monomers are shown in Fig. 2b. The characteristic peaks of the methine groups in Hantzsch rings (4.96–3.79 p.p.m.) can be clearly identified. The integral ratio between the protons in the vinyl and methine groups in 1,4-DHP rings ($I_{6.08–5.93}/I_{5.73–5.61}/I_{4.96–3.79}$) is 1 : 1 : 0.97–1.04, which is consistent with the theoretical value (1 : 1 : 1). Similar results were obtained when other 1,3-cyclohexanedione derivatives were used (Supplementary Figs. 1–4). These results suggest the facile preparation of different Hantzsch monomers via the HTP Hantzsch reaction.

These M(X)(Y) monomers were copolymerized with commercially available poly(ethylene glycol) methyl ether methacrylate (PEGMA, $M_n$: ~950 g mol$^{-1}$) via convenient radical polymerization to obtain water-soluble copolymers in an HTP manner (Fig. 2a). All polymers had high monomer conversions (93–99%; Supplementary Table 1) and satisfactory molecular weights ($M_n$(GPC): 38,600–186,000 g mol$^{-1}$; Supplementary Table 1 and Supplementary Fig. 5, P(X)(1) as a typical example). These results suggest that different 1,4-DHP groups in M(X)(Y) are compatible with radical polymerization. It is noticed that P(4)(Y) have broader polydispersity indices (PDIs) (3.56–9.23) than other polymers. This might be attributed to the N,N-dimethylaniline moieties in P(4)(Y), which possibly produce radical during polymerization[33,34]. The N,N-dimethylaniline radical in P(4)(Y) might link other polymer chains leading to broad PDIs.

Polymers P(X)(Y) were obtained by simple precipitation in diethyl ether. In a typical example, the $^1$H NMR spectra of P(X)(1) (Fig. 2c) showed characteristic peaks of 1,4-DHP moieties (4.94–3.79 p.p.m.) and methoxy groups in P(PEGMA) segments (3.20 p.p.m.). The integral ratios between the 1,4-DHP methines and methyl groups in PEG chain ends ($I_{4.94–3.79}/I_{3.20} = 0.95–1.05$: 3; Supplementary Table 1) matched well the theoretical values (1 : 3). Other polymers showed similar results (Supplementary Figs. 1–4 and Supplementary Table 1), which suggests the

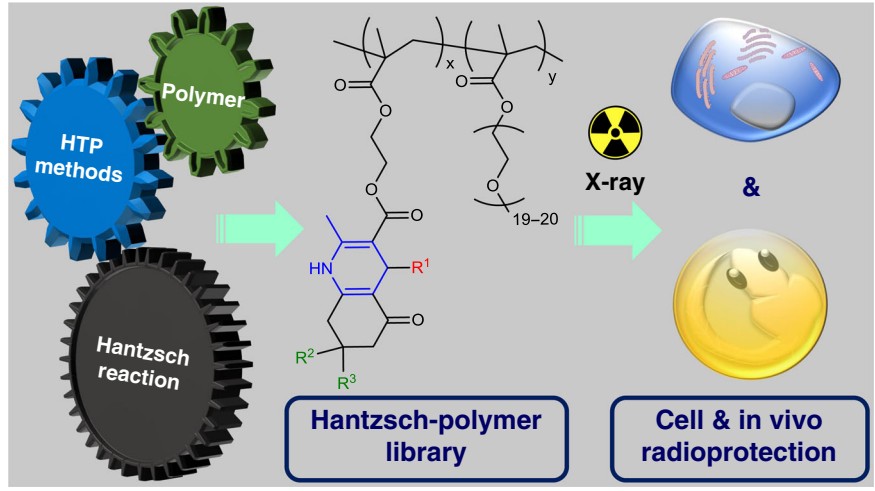

**Fig. 1 Anti-radiation polymers based on Hantzsch's reaction.** Exploration of anti-radiation polymers by combining Hantzsch's reaction, HTP technology and polymer chemistry.

**Fig. 2 The Hantzsch monomers and related polymers. a** HTP preparation of 45 monomers via Hantzsch's reaction and 45 polymers via free radical polymerization. **b** [1]H NMR spectra (DMSO-$d_6$, 400 M) of M(X)(1) monomers. **c** [1]H NMR spectra (DMSO-$d_6$, 400 M) of P(X)(1) polymers.

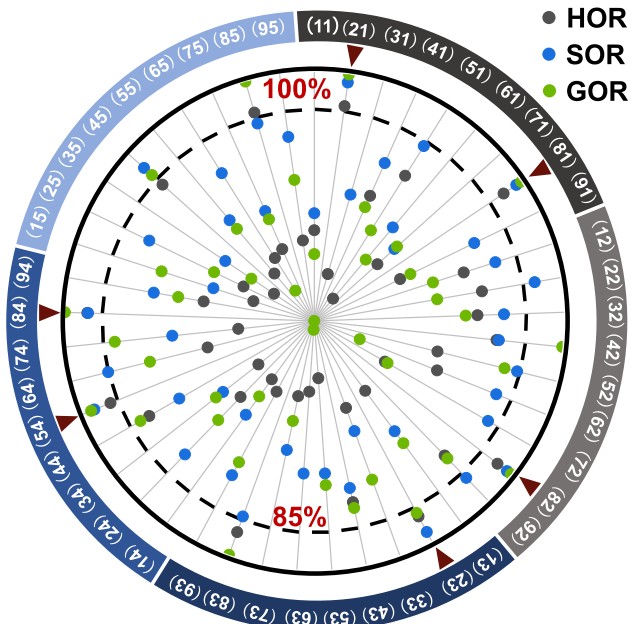

**Fig. 3 Anti-HOR, anti-SOR and anti-GOR ability of polymers.** Polymers scavenging over 85% of three radicals were selected for the next study (red arrows). Source data are provided as a Source Data file.

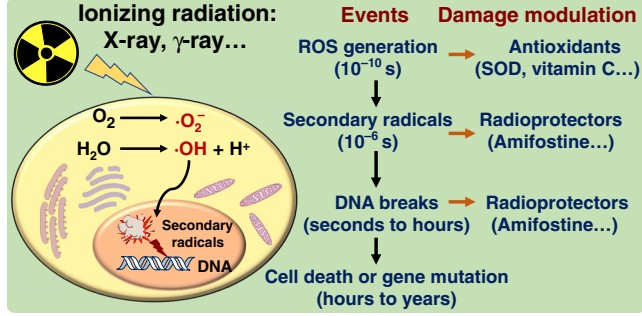

**Fig. 4 Radiation and radioprotectors.** Events after radiation exposure and possible damage modulation by radioprotectors.

successful preparation of the desired copolymers after the HTP radical polymerization.

**First round of screening: HTP measurements of the radical-scavenging ability of P(X)(Y).** Radioprotectors frequently originate from radical scavengers[3]. Hydroxyl radical ($\bullet$OH, HOR) and superoxide radical ($\bullet O_2^-$, SOR) are typical reactive oxygen species (ROS) that are generated by exposing the water and oxygen, respectively, in the cells to ionizing radiation. Galvinoxyl radical (GOR) is an oxygen radical that is commonly used in studies on anti-oxidants[35]. Thus, the scavenging ability of P(X)(Y) for HOR, SOR and GOR was measured according to previous studies[35–37] in an HTP manner (Supplementary Table 2).

The ability of polymers to scavenge different radicals is listed in the radar chart in Fig. 3, which clearly shows that the radical-scavenging ability of polymers depends on the radical species and combinations of A(X) and B(Y). This suggests the rationality behind and necessity of building a sample library to exploit a radioprotector. Six polymers, i.e., P(2)(1), P(2)(3), P(5)(4), P(8)(1), P(8)(2) and P(8)(4) (Fig. 3, red arrows), were selected for the next study, because they effectively scavenged over 85% of all three radicals.

**Second round of screening: cellular experiment for the radioprotective ability of selected polymers.** Ionizing radiation rapidly produces many ROS in living organisms. These ROS quickly generate diffusible secondary radicals that attack the DNA and lead to DNA breaks as well as cell and organ damage[5] (Fig. 4). Many natural anti-oxidants, such as vitamins and polyphenols, have been verified to be poor radioprotectors despite their excellent radical-scavenging ability[38]. This may be attributed to their poor performance in scavenging the quickly generated secondary radicals[5]. Thus, cellular experiments are necessary to identify radioprotective agents from anti-oxidants. Here, the murine fibroblast cell line L929 was used as the model cell to test the radioprotective ability of six selected polymers.

The cytotoxicity of the polymers was evaluated prior to the radioprotection experiment using a Cell Counting Kit-8 (CCK-8)

assay (Supplementary Fig. 6). The cytosafe concentration of polymers was defined as that at which more than 90% of the cells remained viable. Hence, the polymers were used at their cytosafe concentrations (P(2)(1): 10 mg/mL; P(2)(3): 2 mg/mL; P(5)(4): 0.4 mg/mL; P(8)(1): 1 mg/mL; P(8)(2): 2 mg/mL; P(8)(4): 2 mg/mL) for cell protection against high doses of radiation. Cells were exposed to X-ray radiation (RS-2000 Pro; Radsource, USA) until the cumulative radiation dose reached 80 Gy (7.6 Gy/min), then cultured for 48 h prior to analyses. Amifostine (0.3 mg/mL; Supplementary Fig. 7) and a homopolymer P(PEGMA) prepared by radical polymerization (10 mg/mL, $M_n$(GPC): 76,000 g mol$^{-1}$, Supplementary Fig. 8a and cell viability: 95.2% at 10 mg/mL, Supplementary Fig. 8b) were used as the controls. The cells in the culture medium only served as a blank.

A fluorescein diacetate/propidium iodide (FDA/PI) double-staining assay was used to simultaneously observe living and dead cells using laser scanning confocal microscopy (LSCM) (Fig. 5a). Almost no cells survived 80 Gy X-ray radiation in the blank and P(PEGMA) groups, which suggests that high doses of X-ray are lethal to cells and P(PEGMA) had almost no radioprotective ability. The selected polymers demonstrated a concentration-dependent radioprotective ability. P(5)(4) (0.4 mg/mL) conferred no protection to the cells. Few cells survived radiation in the presence of P(8)(1) (1 mg/mL). P(2)(3), P(8)(2) and P(8)(4) at 2 mg/mL protected the cells better than P(5)(4) and P(8)(1) did. Cells cultured with amifostine (0.3 mg/mL) showed better viability compared to those with P(2)(3), P(8)(2) and P(8)(4), and only few dead cells (red spots) were observed, which confirms the excellent radioprotective capability of amifostine. However, nearly all cells survived the fatal X-ray radiation with 10 mg/mL P(2)(1). The results of direct staining confirmed the results of the quantitative analyses obtained via the CCK-8 assay (Supplementary Fig. 9) and colony formation assay (Supplementary Fig. 10). According to cell viability at different doses of X-ray radiation (Supplementary Fig. 11a), the cellular dose-reduction factors (DRFs(cell)) of amifostine (0.3 mg/mL) and P(2)(1) (10 mg/mL) were calculated as 3.7 and 15.3, respectively (Supplementary Fig. 11b). Flow cytometry was used as previous literatures[39,40] to analyse cell necrosis after exposure to 80 Gy X-ray radiation (Fig. 5b) and the gating strategy (Supplementary Fig. 12) was used according to reported literatures[41]. Cells cultured with P(2)(1) had the lowest level of cell necrosis (~7.8%) as compared to those cultured with other polymers (~30.4–61.7%) and amifostine (~21.9%). This result is similar to that of cells cultured in a medium only or with P(2)(1) (10 mg/mL) for 48 h without exposure to radiation (Supplementary Fig. 13a, cells cultured in a medium only: ~7.1%; Supplementary Fig. 13b, cells cultured with P(2)(1) (10 mg/mL): ~7.9%). When WR-1065 (the active form of amifostine in vivo) (0.3 mg/mL, Supplementary Fig. 14a) was tested, it had slightly weaker radioprotection effect than

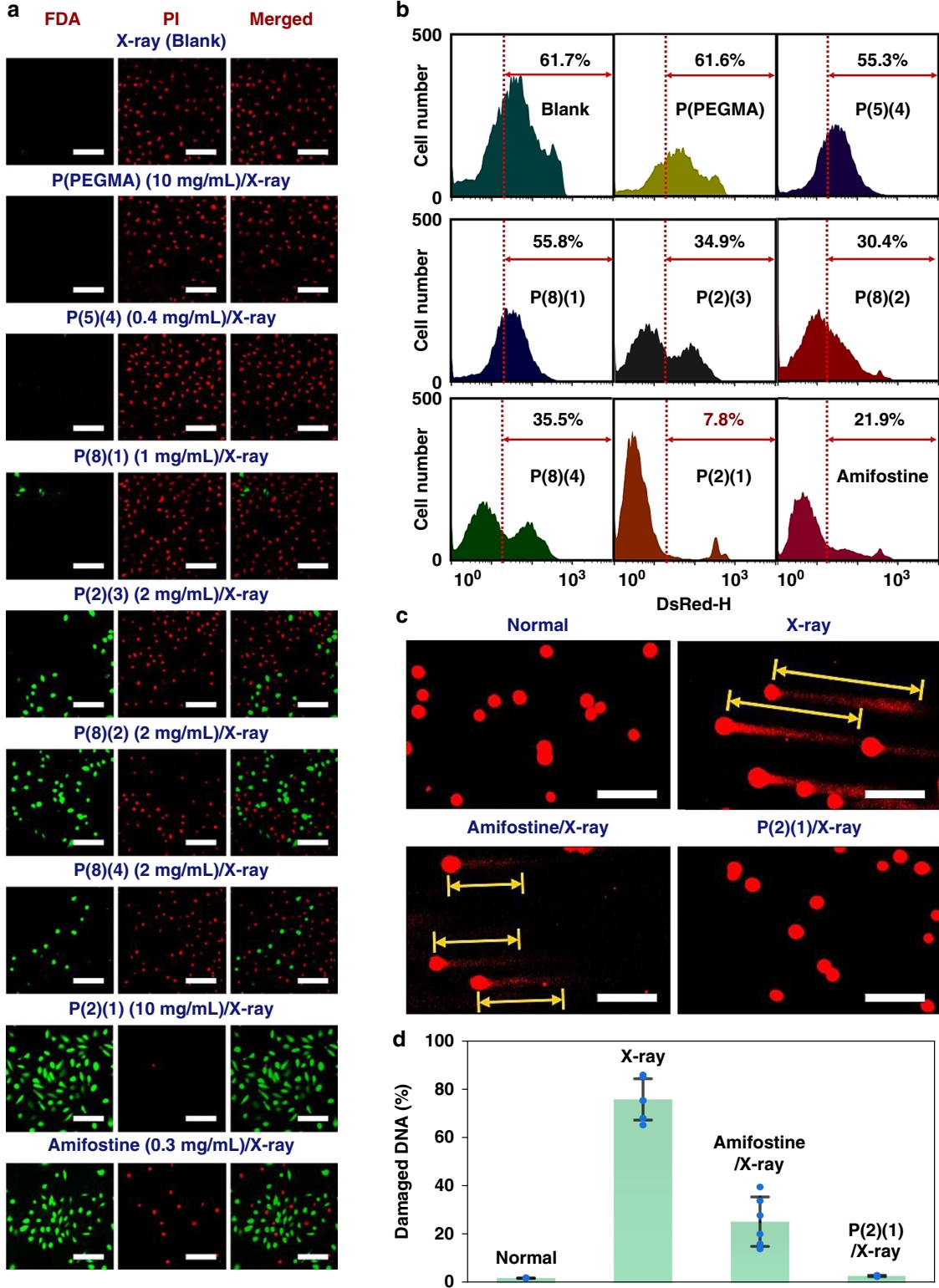

**Fig. 5 Cell experiments to evaluate radioprotection ability of different compounds. a** FDA/PI double staining of L929 cells after exposure to 80 Gy X-ray radiation under different culture conditions. Scale bar = 100 μm. This experiment was repeated three times independently with similar results. **b** Flow cytometry analysis of cell necrosis under different conditions. **c** Comet assay images of cells under different culture conditions. Scale bar = 200 μm. **d** Damaged DNA (%) in the cells. The data are presented as mean values ± SD ($n = 6$ biologically independent cells). Source data are provided as a Source Data file.

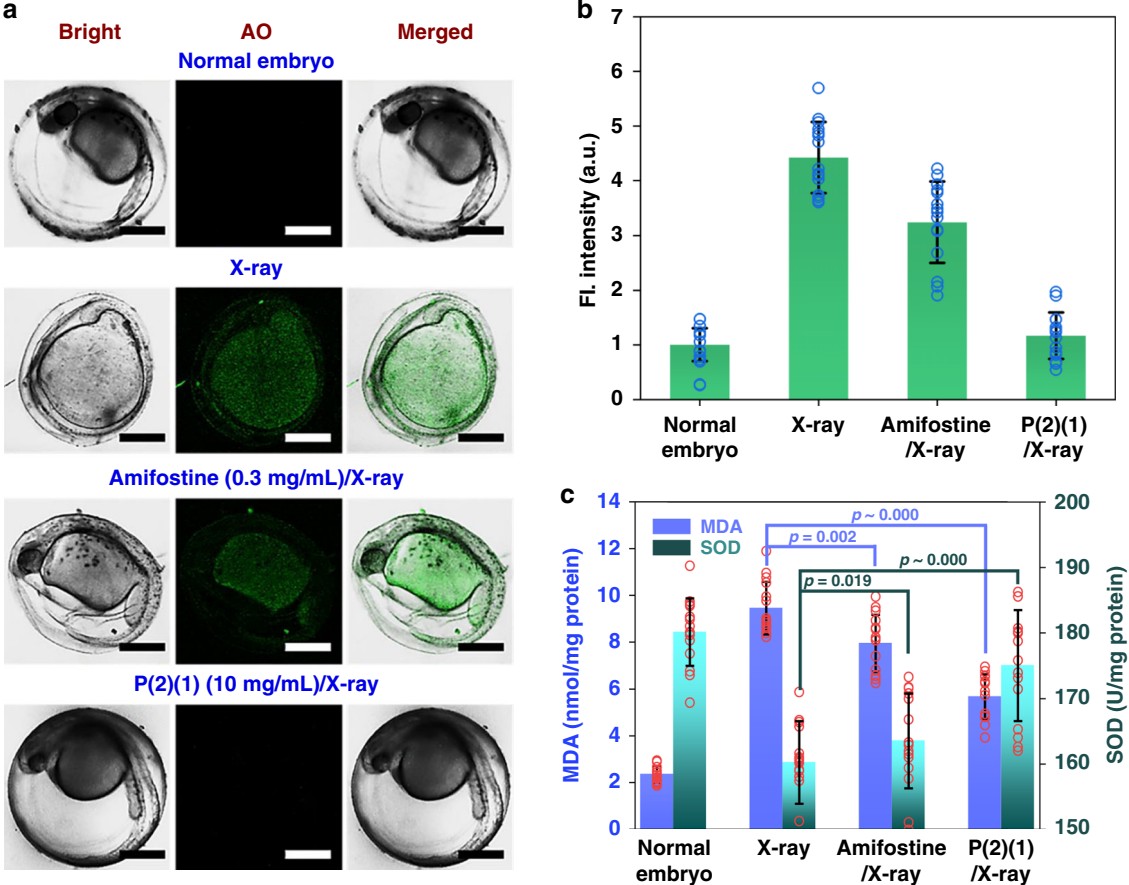

**Fig. 6 In vivo experiments (zebrafish embryos) to test radioprotection ability of different compounds. a** LSCM images of zebrafish embryos (60 hpf) under different conditions. Scale bar = 300 μm. **b** Fluorescence intensity (AO) in zebrafish embryos under different conditions. Data are presented as mean values ± SD ($n = 15$ independent animals). **c** SOD and MDA levels in the zebrafish embryos under different conditions. Data are presented as mean values ± SD ($n = 15$ independent animals). Statistical analyses of results were performed by Student's $t$-test for independent samples (one-side), there was no adjustment made for multiple comparisons. Source data are provided as a Source Data file.

amifostine on L929 cells (Supplementary Fig. 14b, c). These results suggest the feasibility of exploring radioprotective polymers using Hantzsch's reaction.

Subsequently, the radioprotection mechanism was investigated. Radiation-induced ROS in cells are the direct cause of a series of serious consequences[5]. Thus, the ROS levels in cells under different culture conditions were detected by using 2,7-dichlor-odihydrofluorescein diacetate (DCFH-DA) probe as previously described[42]. Fluorescence was hardly detected in normal cells during a 48 h culture (Supplementary Fig. 15a, a') but obvious and increased with time in cells after exposure to 80 Gy X-ray radiation (Supplementary Fig. 15b, b'). Both amifostine (0.3 mg/mL) and P(2)(1) (10 mg/mL) effectively controlled the ROS levels in cells; cells with amifostine had brighter fluorescence (higher ROS levels) than those with P(2)(1) (Supplementary Fig. 15, amifostine: (c, c'); P(2)(1): (d, d')). These results agreed well with the quantitative measurements of ROS in cells (Supplementary Fig. 15e), indicating the anti-radiation ability of amifostine and P(2)(1) is correlated with their ability to scavenge radiation-induced ROS in cells.

Moreover, DNA is the primary target of radiation damage, clustered DNA damage with lesions in 10–20 basepair is the hallmark of ionizing radiation[43]. Thus, single-cell gel electro-phoresis (comet assay) was performed according to the literature[44,45] to detect damaged DNA in cells under different conditions (Fig. 5c). Compared to normal cells, the cells that were exposed to 80 Gy X-ray radiation clearly had long tails (damaged

DNA), which suggests that this X-ray dose can easily destroy cellular DNA. Amifostine (0.3 mg/mL) partially protected the DNA, leading to shorter tails compared to the X-ray group. Cells cultured with P(2)(1) (10 mg/mL) showed negligible tails and clear blue fluorescence owing to the 1,4-DHP group in P(2)(1) (Supplementary Fig. 16), which suggests that the P(2)(1) in the cells effectively protected the DNA against high doses of X-ray radiation. These results are consistent with the quantitative analyses of damaged DNA in the cells (Fig. 5d), which confirms that both P(2)(1) and amifostine are efficient radioprotectors because they can effectively protect cellular DNA from radiation damage; large doses of P(2)(1) can be used because of its excellent cytosafety, which offers better cellular protection compared to amifostine and other polymers. Therefore, P(2)(1) was selected from the six polymers for the next in vivo study.

**Radioprotection of zebrafish embryos against high doses (80 Gy) of X-ray.** The genes of zebrafish exhibit >85% similarity to those of humans. Zebrafish embryos have been used as a unique vertebrate model to quickly screen therapeutic agents[46,47]. Zebrafish embryos are small, optically transparent and easily available, and have short embryonal development, which are conducive to quick and direct observation of radiation-induced damage to animals (death, deformation, abnormal organs and damaged DNA); thus, zebrafish embryos have also been used to study radiation-induced mutation and evaluate radioprotective agents including amifostine[48–53]. Here we used zebrafish embryos

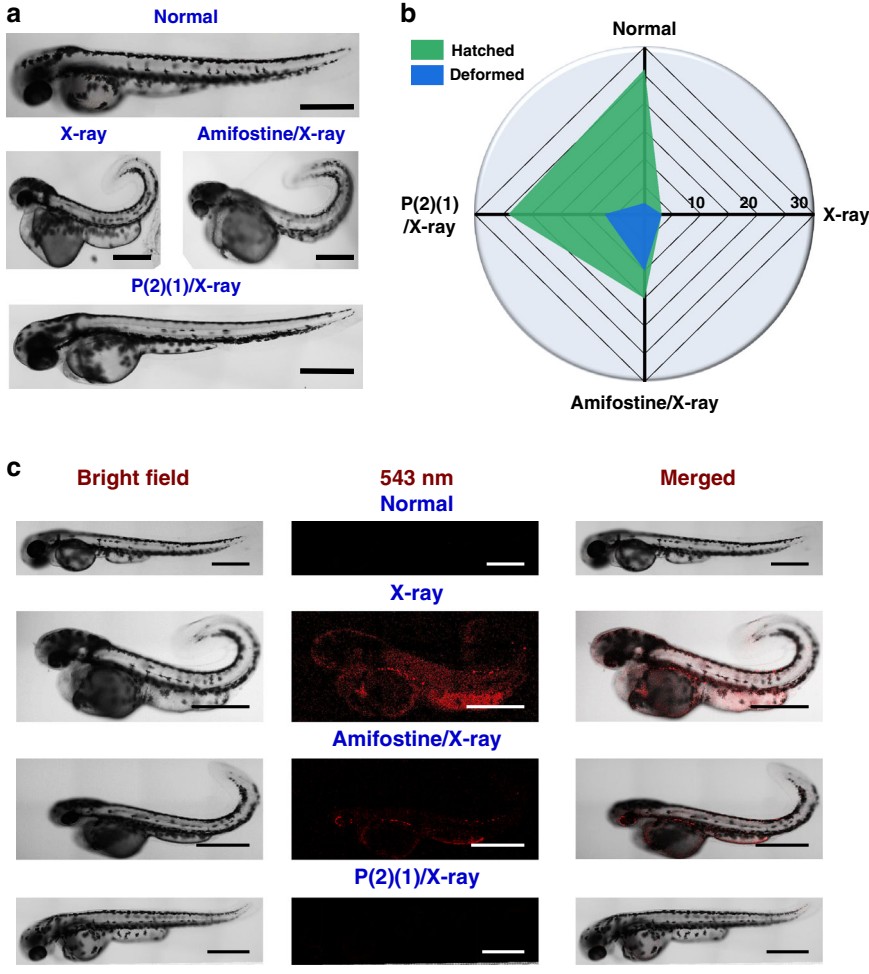

**Fig. 7 Radiation sequela of zebrafish embryos in the presence of different compounds. a** Representative pictures of zebrafish larvae hatched from embryos under different conditions. Seven days, scale bar = 500 μm. **b** Hatched larvae number and deformed larvae number, 30 samples/group. **c** Damaged DNA in zebrafish larvae hatched from embryos under different conditions. Scale bar = 500 μm. This experiment was repeated twice independently with similar results. Source data are provided as a Source Data file.

as the model to study the in vivo protection by P(2)(1) against a high dose (80 Gy) of X-ray radiation.

A total of 60 zebrafish embryos (12 h post-fertilization, hpf) were placed in a 35 mm glass culture dish with P(2)(1) (10 mg/mL, Holtfreter's solution) and exposed to X-ray radiation (cumulative radiation dose: 80 Gy, 7.6 Gy/min). Then, these embryos were cultured (28.5 °C) for 48 h prior to analysis. Embryos cultured in Holtfreter's solution and amifostine (0.3 mg/mL, Holtfreter's solution) were used as the blank and control, respectively.

Embryos (15 samples/group, randomly selected) were stained with Acridine Orange (AO), which is a metachromatic dye, to selectively stain nucleic acids in cells that underwent apoptosis/necrosis. Images of the embryos were captured using LSCM (Fig. 6a) and the fluorescence intensity was analysed using ImageJ software to measure cells with necrosis (Fig. 6b).

Normal embryos (~60 hpf) developed recognizable eyes and tails. Embryos exposed to X-ray lost these organs and showed clearly visible green fluorescence, which indicates embryonic death induced by radiation. Embryos cultured with amifostine showed detectable eyes and tails. However, abdominal swelling and pronounced fluorescence were also observed, which suggests the inadequate protection of zebrafish embryos by amifostine. Conversely, embryos in the presence of P(2)(1) had indiscernible fluorescence of AO. These embryos developed eyes and tails

similarly to normal embryos, which suggests that P(2)(1) plays an anti-radiation role in vivo. The visible results are consistent with the AO fluorescence intensity analyses (Fig. 6b).

Decreased activity of superoxide dismutase (SOD, a radical scavenger) and generation of malondialdehyde (MDA, a lipid peroxidation end product) in organisms are biomarkers of radiation-induced damage[53–55]. Thus, the levels of SOD and MDA in the embryos were measured (Fig. 6c; 15 samples/group, randomly selected). The MDA and SOD levels in normal embryos were 2.35 nmol/mg protein and 180.12 U/mg protein, respectively. The X-ray radiation considerably increased the MDA level and decreased the SOD level in the embryos (MDA: 9.45, SOD: 160.25). P(2)(1) considerably improved these abnormal indicators (MDA: 5.68, $p \sim 0.000$; SOD: 175.03, $p \sim 0.000$, compared with the X-ray group), whereas amifostine weakly affected these indicators (MDA: 7.95, $p = 0.002$; SOD: 163.54, $p = 0.019$, compared with the X-ray group). These results suggest that P(2)(1) is better than amifostine in terms of protecting zebrafish embryos from deadly radiation.

**Radiation sequela of zebrafish embryos**. Typically, zebrafish embryos form larvae by ~120 hpf[56]. Thus, embryos (30 samples/group) were continually cultured for 7 days to investigate the influence of 80 Gy X-ray radiation exposure on embryogenesis. The morphology of the larvae from embryos under different

conditions was photographically recorded (Fig. 7a and Supplementary Fig. 17). The number of hatched larvae was also recorded (Fig. 7b). Larvae with curved spines were considered deformed[57] (Fig. 7a, b and Supplementary Fig. 17).

Thirty normal embryos hatched 26 larvae (Fig. 7a and Supplementary Fig. 17a; hatching rate: 87%), including two deformed larvae (Fig. 7b; deformation rate: 8%). After exposure to 80 Gy X-ray radiation, only three embryos hatched larvae (hatching rate: 10%), but they were all deformed (Fig. 7a, b and Supplementary Fig. 17b; deformation rate: 100%). This indicates considerable damage caused by X-ray on the development of zebrafish embryos. Amifostine (0.3 mg/mL) improved the hatching rate (50%) of the embryos and resulted in 15 larvae, among which 10 were deformed (Fig. 7a, b and Supplementary Fig. 17c; deformation rate: 67%). Embryos cultured with P(2)(1) smoothly hatched 24 larvae (hatching rate: 80%). Among these larvae, 17 had a normal straight spine (Fig. 7a, b and Supplementary Fig. 17d; deformation rate: 29%). According to the relationship between X-ray doses and the hatched/deformed larvae numbers (Supplementary Fig. 18), the DRF(fish, hatched) and DRF(fish, deformed) values of P(2)(1) (10 mg/mL) were calculated as 10.9 and 11.6, respectively; these values of amifostine (0.3 mg/mL) were 2.3 and 3.7, respectively. These results suggest that P(2)(1) is superior to amifostine in terms of preventing radiation-induced animal death and distortion.

Radiation-induced DNA damage leads to severe biological consequences such as chromosome aberration, cell death and transformation[52,58]; thus, the damaged DNA in the hatched larvae was assessed. All larvae were kept in a primary antibody solution (Phospho-Histone H2AX (Ser139) Rabbit Monoclonal Antibody; 1 : 500 dilution) for 1 h and then incubated with an Alexa Fluor 555-labelled secondary antibody (Alexa Fluor 555-labelled donkey anti-rabbit IgG, 1 : 500 dilution) for another 1 h prior to observation by LSCM (Fig. 7c).

Compared with the normal larvae, the larvae hatched from embryos after X-ray exposure showed pronounced red fluorescence, which indicated impaired DNA in the larvae. Embryos cultured with amifostine also generated larvae with red fluorescence. Larvae from embryos cultured with P(2)(1) had undetectable fluorescence. These results suggest that P(2)(1) efficiently counteracted the radiation-induced DNA damage and acted as an anti-radiation agent.

**Challenges and possible extensions.** We constructed a small model library of polymers according to the HTP strategy to select a simple polymer (random polymer chain and a broad PDI) as a promising radioprotector. However, 45 samples do not represent real HTP; although phenol and ferrocene substituents have been identified in improving the antioxidant capability of polymers, no rule has been summarized to guide the future development of polymer anti-radiators in a more efficient way. In future research, larger sample libraries should be prepared and theoretical calculation should be used to summarize rules for de novo design of polymers with better anti-radiation capability and biosafety.

Recent studies have shown that polymer structures (i.e., monomer sequences and topology structures) considerably affect the performance when polymers are used as biomaterials[59–67]. Thus, preparing well-defined polymers will improve the current research. Currently, polymers with precise structures can be rapidly prepared via modern control radical polymerization (CRP) technologies. Examples include single electron-transfer–atom transfer radical polymerization (ATRP), photoinduced ATRP, photo-induced electron/energy transfer–reversible addition-fragmentation chain transfer (RAFT) and sulphur-free RAFT emulsion polymerization[66,68–76]. Future

applications of these modern CRP techniques with the methods developed in the current research may result in polymer libraries with more samples and higher molecular diversity that will include different side groups, controlled monomer sequences and molecular weights. This development will accelerate the identification of polymers with enhanced anti-radiation ability and other bioactivities. Meanwhile, polymers, as potential therapeutic agents, should be biodegradable. However, in this research, polymers were not biodegradable. In future studies, combining other polymerization methods (e.g., poly-condensation and ring-opening polymerization) and radioprotective groups may result in biodegradable radioprotective polymers for in vivo applications.

Mammals are better than zebrafish as animal models for pharmaceutical research, because they are closer to human and more suitable to simulate clinical treatments. In future research of polymers with improved radioprotective ability and biocompatibility, mice and rats can be used as animal models to study not only anti-radiation capability but also pharmaceutical parameters (e.g., administration route/frequency, pharmacokinetics, pharmacodynamics and drug distribution) of these polymers.

In summary, we prepared 45 monomers using Hantzsch's reaction in an HTP manner. These monomers were copolymerized with PEGMA to produce 45 water-soluble polymers via HTP radical polymerization. These polymers were then screened stepwise according to different criteria via HTP measurements, to finally achieve a biocompatible polymer that can effectively protect cells and zebrafish embryos from lethal doses of X-ray radiation. The protective effect of the developed polymer was better than that of amifostine. This highlights the value of using MCR and HTP technologies in polymer chemistry to identify functional polymers for potential applications.

This study, hence, allows developing safe and efficient polymeric radioprotectors. In addition, it offers insights into developing functional polymers via MCRs. This study may, therefore, prompt a broad study of MCRs and HTP methods in polymer science and lead to the development of other functional polymers for inter-disciplinary applications.

## Methods

**HTP measurements of the radical-scavenging ability of P(X)(Y).** The ability of polymers to scavenge different radicals (HOR, SOR and GOR) was measured according to literatures[35–37] in an HTP manner.

*Anti-HOR assay.* Phosphate-buffered saline (PBS) solutions (pH ~ 7.4) of salicylic acid (20 μL, 10 mM), FeSO$_4$ (20 μL, 10 mM), H$_2$O$_2$ (20 μL, 10 mM) and polymers (100 μL, 10 mg/mL) were mixed in a 96-well plate. The mixtures were incubated at 37 °C for 15 min; then, the characteristic absorption of 2,3-dihydroxybenzoic acid, which is converted from salicylic acid by •OH, was recorded (510 nm). A mixture of salicylic acid (20 μL, 10 mM), FeSO$_4$ (20 μL, 10 mM), H$_2$O$_2$ (20 μL, 10 mM), and PBS (100 μL, pH ~ 7.4) was used as a blank (anti-HOR ability: 0%). Mixtures of salicylic acid (20 μL, 10 mM), FeSO$_4$ (20 μL, 10 mM), PBS (20 μL, pH ~ 7.4) and polymers (100 μL, 10 mg/mL) were defined as 100% anti-HOR ability, respectively. The inhibition efficiency of 2,3-dihydroxybenzoic acid by polymers reflects the anti-HOR ability of polymers. Each sample was tested five times in parallel.

*Anti-SOR assay.* PBS solutions (pH ~ 7.4) of xanthine (XAN, 10 μL, 0.4 mM), xanthine oxidase (XOD, 10 μL, 0.05 u/mL), nitro-blue tetrazolium (NBT, 10 μL, 0.24 mM) and polymers (100 μL, 10 mg/mL) were mixed in a 96-well plate. The mixtures were incubated at 37 °C for 15 min; then, the characteristic absorption of methyl hydrazone that is converted from NBT by •O$_2^-$ was recorded (560 nm). A mixture of XAN (10 μL, 0.4 mM), XOD (10 μL, 0.05 u/mL), NBT (10 μL, 0.24 mM) and PBS (100 μL, pH ~ 7.4) was used as a blank (anti-SOR ability: 0%). Mixtures of XAN (10 μL, 0.4 mM), PBS (10 μL, pH ~ 7.4), NBT (10 μL, 0.24 mM) and polymers (100 μL, 10 mg/mL) were defined as 100% anti-SOR ability, respectively. The inhibition efficiency of methyl hydrazone by polymers reflects the anti-SOR ability of polymers. Each sample was tested five times in parallel.

*Anti-GOR assay.* Polymer solutions (100 μL, 10 mg/mL in PBS (pH 7.4)) were added in a 96-well plate; then, a solution of GOR (100 μL, 0.2 mg/mL in ethanol) was added to each polymer solution. The mixtures were incubated at 37 °C for

30 min and the absorbance values were recorded (450 nm). A PBS solution (100 μL, pH 7.4) was used as a blank (anti-GOR ability: 0%). Polymers (100 μL, 10 mg/mL in PBS (pH 7.4)) were mixed with ethanol (100 μL) and defined as 100% anti-GOR ability, respectively. Each sample was tested five times in parallel.

**Cell culture**. L929 cells (a fibroblast cell line from mice) were cultured in a Roswell Park Memorial Institute-1640 medium supplemented with 10% fetal bovine serum (FBS) and 1% penicillin and streptomycin. The cells were then incubated at 37 °C in 5% $CO_2$. The culture medium was changed every two days to maintain the exponential growth of cells.

**Cytotoxicity evaluation**. The cytotoxicity of different samples (P(X)(Y), P (PEGMA), amifostine and WR-1065) to L929 cells was evaluated by a CCK-8 assay. Briefly, cells ($\sim 5 \times 10^4$ cells/mL) were seeded in a 96-well plate in culture medium (100 μL, 10% FBS and 1% penicillin and streptomycin). After attachment, cells were washed with PBS and added culture medium containing different concentrations of samples. After a 48 h culture, cells were washed with PBS three times, then incubated in 100 μL of culture medium containing 10% CCK-8 solution (37 °C, 2 h). The plate was put into a microplate reader (VICTOR$^{TM}$ X3 PerkinElmer 2030 Multilabel Plate Reader) to record the absorbance (450 nm). The absorbance of cells in culture medium only was defined as 100% viability. The absorbance of a culture medium (without cells) was defined as 0%. Data were present as mean ± SD ($n = 10$) to indicate the cytotoxicity of different samples to L929 cells.

Cytosafe concentrations of different samples were defined as those where cells remained more than 90% viability.

**Cellular experiment for radioprotection ability**. L929 cells ($\sim 5 \times 10^4$ cells/mL) cells were incubated with P(2)(1) (10 mg/mL, in culture medium) for 0.5 h followed by exposure to X-ray irradiation (Radsource, RS-2000 pro) until the accumulative radiation dose reached 80 Gy (7.6 Gy/min). Then, the cells were cultured with P(2)(1) (10 mg/mL, in culture medium) for 48 h and added the PBS-FDA-PI mixed solution (FDA: 3 μg/mL; PI: 3 μg/mL) to simultaneously observe the live and dead cells through 450–490 nm and 515–560 nm band-pass excitation filters (I3 and N2.1) by a fluorescence microscope. Other polymers and amifostine were parallelly tested at their cytosafe concentrations. Cells in culture medium only served as a blank.

For CCK-8 assay, a CCK-8 solution instead of PBS-FDA-PI mixed solution was used to quantitatively evaluate the cell viability.

For flow cytometry analysis, cells exposed to X-ray and cultured with different compounds for 48 h; then, a PBS solution of PI (10 μg/mL) was added and scattered to cells for 15 min. The cell apoptosis rate was evaluated by a flow cytometry (BD Calibur, $\lambda_{ex} = 488$ nm) and the gating strategy was used according to reported literatures[41]. Cells in culture medium only and amifostine (0.3 mg/mL) were used as a blank and a control, respectively.

The WR-1065 (0.3 mg/mL) was similarly measured.

**Colony formation assay**. L929 cells ($\sim 100$ cells/well in 24-well plates) were incubated with P(2)(1) (10 mg/mL, in culture medium) for 0.5 h followed by exposure to X-ray irradiation until the accumulative radiation dose reached 80 Gy (7.6 Gy/min). Then, the cells were cultured with P(2)(1) (10 mg/mL, in culture medium) at 37 °C with 5% $CO_2$ for 48 h. These cells were continually cultured for another 12 days with normal culture medium, the colonies were fixed with paraformaldehyde (4% in $H_2O$) for 20 min (25 °C) and stained with crystal violet aqueous solution (0.2%) for 10 min (25 °C) followed by washing with PBS twice. The colonies containing more than 50 cells were recorded as survivors; data were present as mean ± SD ($n = 3$) and the colony number of cells without X-ray was defined as 100%. Other polymers and amifostine were parallelly tested at their cytosafe concentrations. Cells in culture medium only served as a blank group.

**ROS induced by X-ray**. The ROS levels were measured using a DCFH-DA probe as previously described[42]. Briefly, L929 cells ($\sim 5 \times 10^4$ cells/mL) were incubated with P(2)(1) (10 mg/mL, in culture medium) for 0.5 h followed by exposure to X-ray irradiation until the accumulative radiation dose reached 80 Gy (7.6 Gy/min). Culture medium was removed at different time points (0, 3, 6, 12, 24, 48 h), cells were washed with a serum-free medium and cultured in a working solution containing DCFH-DA (10 μmol L$^{-1}$, 100 μL) at 37 °C for 20 min. The changes of the fluorescein in cells were collected (485 nm/535 nm) by a microplate reader (VICTOR™ X3 PerkinElmer 2030 Multilabel Plate Reader) and observed by LSCM ($\lambda_{ex} = 488$ nm). The fluorescence intensity of cells at 0 h was defined as the base line (1.0); the increased ROS levels were shown as relative fluorescence intensity compared with the base line and presented as mean ± SD ($n = 5$). Cells in culture medium only served as a blank group, cells in culture medium without X-ray as a normal group.

ROS levels in cells cultured with amifostine (0.3 mg/mL) were similarly measured.

**Single-cell gel electrophoresis (comet assay)**. Alkaline comet experiment was used to detect damaged DNA in cells after X-ray irradiation according to the instruction book (Trevigen).

L929 cells ($\sim 5 \times 10^4$ cells/mL) were incubated with P(2)(1) (10 mg/mL) for 0.5 h followed by exposure to X-ray irradiation (Radsource, RS-2000 pro) until the accumulative radiation dose reached 80 Gy (7.6 Gy/min). Cells were collected and added lysis solution (1 mL) followed by keeping at 4 °C for 20 min. Cell lysis solution (5 μL) was mixed with LMAgarose (1%, 37 °C, 45 μL), then dropped on a comet slide (Trevigen) to generate a gel at 25 °C in 10 min. The comet slide was steeped in the lysis solution for 60 min, then kept in an alkaline electrophoresis solution (10 mL, NaOH: 8 g/L, EDTA: 500 mM, in dH$_2$O) for another 60 min (25 °C, dark). Then, the comet slide was placed in a comet assay electrophoresis slide tank containing prechilled alkaline electrophoresis solution (950 mL). After running at 21 volts for 10 min, the comet slide was taken out and washed twice with $H_2O$ then dehydrated in 70% ethanol (10 min). This comet slide was added SYBR Gold (100 μL, 1/10000 diluted) and kept in a refrigerator (4 °C) for 5 min, then washed twice with $H_2O$ prior to observation of damaged DNA ($\lambda_{ex} = 543$ nm) and fluorescence of P(2)(1) ($\lambda_{ex} = 405$ nm), respectively, by a LSCM.

Cells in medium only and amifostine (0.3 mg/mL) were used as a blank and a control, respectively. Normal cells (without irradiation) were analysed through the same process.

**Maintenance of fish and egg spawning**. All zebrafish used in this study were the Tuebingen strain. Adult fish were fed with live adult brine shrimp in the morning and evening, and the adult zebrafish used to produce eggs in this study were kept in a water-circulating system at 28.5 °C. Fertilized eggs were raised at 28.5 °C in Holtfreter's solution (0.059 M NaCl, 0.00067 M KCl, 0.00076 M CaCl$_2$ and 0.0024 M NaHCO$_3$). The embryos ($\sim 12$ hpf) were used for the experiments. Ethical approval was obtained from the Animal Care and Use Committee of Tsinghua University. All experimental animal procedures were performed under anaesthesia and all efforts were made to minimize suffering.

**Radioprotection of zebrafish embryos against X-ray**. Sixty zebrafish embryos (12 hpf) were placed in a 35 mm glass culture dishes with P(2)(1) (10 mg/mL, in Holtfreter's solution) and exposed to X-ray irradiation (accumulative radiation dose: 80 Gy, 7.6 Gy/min). These embryos were cultured (28.5 °C) for 48 h prior to analyses. Embryos cultured in Holtfreter's solution and amifostine (0.3 mg/mL in Holtfreter's solution) were used as a blank and a control, respectively.

Embryos (15 samples/group, randomly selected) were stained by AO (5 μg/mL in $H_2O$, 1 h) and washed twice with $H_2O$. Images of embryos were captured by using a LSCM ($\lambda_{ex} = 488$ nm); the fluorescence intensity was analysed by ImageJ software to measure the necrosis cells.

Embryos (15 samples/group, randomly selected) were homogenized in ice-cold physiological saline (0.5 mL), then centrifuged ($420 \times g$, 20 min, 4 °C). The levels of SOD and MDA were measured according to the protocols of commercially available kits (Jiancheng Institute of Biotechnology, Nanjing, China). SOD activity was expressed as U/mg protein and MDA level was expressed as nmol/mg protein.

**Radiation sequela of zebrafish embryos**. Left embryos (30 samples/group) were continually cultured for 7 days to investigate the influence of 80 Gy X-ray irradiation (7.6 Gy/min) on embryogenesis. Hatched larvae number was recorded. Larvae with curved spines were considered deformed[57,77]. The larvae were paralysed in a tricaine solution (4 μg/mL, ~10 min) prior to observation. The stereomicroscope and LSCM were used for morphological observation.

An immunofluorescence staining was used to detect damaged DNA in the hatched larvae according to reported literatures[78,79]. Larvae were paralysed in a tricaine solution (4 μg/mL, ~10 min), then fixed in 4% paraformaldehyde in PBS (pH 7.2) (4 h, 25 °C). After washing three times by PBS, larvae were sealed in a sealing solution for 1 h, then kept in the primary antibody solution (Phospho-Histone H2AX (Ser139) Rabbit Monoclonal Antibody (1/500 dilution) for 1 h. The larvae were washed three times by PBS, then incubated with the Alexa Fluor 555-labelled secondary antibody (Alexa Fluor 555-Labelled Donkey Anti-Rabbit IgG; 1/500 dilution) for another 1 h. The larvae were washed by PBS three times prior to observation by LSCM ($\lambda_{ex} = 543$ nm).

**DRF of cells and zebrafish**. The DRF was calculated as a ratio of radiation dose required to produce the same biological effect to cells (DRF(cell)) or zebrafish (DRF(fish)) in the presence and absence of the radioprotector as described[80,81].

L929 cells ($\sim 5 \times 10^4$ cells/mL) cells were exposed to different doses of X-ray irradiation (5, 10, 20, 40, 60 and 80 Gy; 7.6 Gy/min) prior to a 48 h culture (37 °C, 5% $CO_2$). CCK-8 assay was used to measure the cell viability, the viability of cells without X-ray was defined as 100%. The DRF(fish) values of different agents (polymers and amifostine at their cytosafe concentrations) were calculated by the ratios between different radiation doses with and without agents to produce same cell viability.

Embryos (30 samples/group) were exposed to different doses of X-ray irradiation (5, 10, 20, 40, 60 and 80 Gy; 7.6 Gy/min). These embryos were cultured (28.5 °C) for 7 days. The hatched larvae number and deformed larvae number were recorded. The DRF(fish, hatched) and DRF(fish, deformed) values of different

agents (P(2)(1) (10 mg/mL) and amifostine (0.3 mg/mL)) were calculated by the ratios between different radiation doses with and without agents to produce animal death and deformation, respectively.

**Statistical analysis**. The results were analysed using SPSS Statistics v.25.0 and MedCalc 18.1 and are presented as mean values ± SD as indicated. Comparisons were performed between two groups using Student's $t$-test (one-side). The exact $p$-values were calculated by SPSS. The sample size was pre-estimated to ensure statistical analysis, and no sample was optionally excluded from the analysis. No blinding was done in the analyses and quantifications.

**Reporting summary**. Further information on research design is available in the Nature Research Reporting Summary linked to this article.

## Data availability

The authors declare that the data supporting the findings of this study are available within the paper and its Supplementary Information files. Source data are provided with this paper.

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

## Acknowledgements

This research was supported by the National Science Foundation of China (21971141 and 21574073) and Tsinghua-UNSW Initiative Scientific Research Program (2020Z02NSW).

## Author contributions

L.T. developed the concept and conceived the experiments. G.L., Y.Z. and T.M. performed the laboratory experiments. S.J. and T.L. provided zebrafish embryos. L.T., G.L. and S.J. contributed to the experimental analyses and wrote the manuscript. L.T., Y.G., S.J. and Y.W. provided financial support to the research.

## Competing interests

The authors declare no competing interests.
