## [Peer Review File · Nature Communications]

Reviewers' comments:

Reviewer #1 (Remarks to the Author):

The authors reported the synthesis of a variety of polymers by combining Hantzsch's reaction and high-throughput methods, after radical scavenge and cytotoxicity evaluating, biocompatible P(2)(1) with good anti-radiation capability was selected as the model to perform in vivo experiments. The results showed that the selected polymer efficiently protected cells and zebrafish embryos from lethal doses of ionising radiation, and it is superior to that of amifostine, which is approved radioprotector for clinical applications. This work possesses the necessary novelty and diversity, the data and figures are of good quality, taking into account everything mentioned above, I recommend this manuscript for publication after the following minor points being addressed.

1. The PDI of the obtained polymers have great disparity (from 1.31 to 9.23), what is the probable reason?
2. What are the criteria for choosing the structure of the monomers in this work? From the results, is there any structure or functional group that helps to improve the anti-radiation capability of the polymer, which is meaningful to develop promising radioprotector in a more efficient way.

Reviewer #2 (Remarks to the Author):

This article deals with the synthesis and evaluation of radioprotective Polymers via Hantzsch's Reaction against In Vivo X-ray damage. This study concerned a recent and novel trend in radioprotection but, the manuscript needs minor revisions

1. Please revise the writing style of the word "ionizing" must be the same all over the manuscript.
- 2-The high light paragraphs in the results and discussion should be preferred to remove to the experimental section
- 3- More discussion and references are needed concerning with the level of cell necrosis , DNA damage , MAD , SOD and zebrafish larvae

Please note Reviewer #2 has also provided a highlighted PDF attached.

Reviewer #3 (Remarks to the Author):

In this study, authors described the preparation and radioprotective effects of new polymers. They used cellular and zebrafish experiments for approving of their radioprotective effect. This study has major defects in technically experiments for evaluation of radioprotective effect. This manuscript is not suitable for publication.

- 1-Which part of polymer is able to scavenge free radicals; mechanisms are unclear?
- 2- Colony assay is necessary to do for evaluation of cellular radioprotective effects.
- 3-Amifostine is a prodrug (thiophosphate part of molecule should be hydrolyzed in vivo to active form) then amifostine is weak in vitro for radioprotective effect, its free thiol derivative should be used.
- 4-Survival of irradiated mice (with and without treatment) should be used as standard in vivo model for demonstrating of radioprotective effect and calculation of DRF for both amifostine and polymers.

Author's Response to Reviewer #1:

1) The PDI of the obtained polymers have great disparity (from 1.31 to 9.23), what is the probable reason?

Answer: Thanks for the professional question. The *N,N*-dimethylaniline moieties in P(4)(Y) might be the reason for high PDIs of P(4)(Y) (from 3.56 to 9.23). During polymerization, *N,N*-dimethylaniline groups possibly produce radical which might link other polymer chains leading to broad PDIs. This information and related references (*Makromol. Chem.* **178**, 3221-3228 (1977); *J. Macromol. Sci. Chem.* **A20**, 789-805 (1983)) have been added in the revised manuscript. Thanks!

2) What are the criteria for choosing the structure of the monomers in this work? From the results, is there any structure or functional group that helps to improve the anti-radiation capability of the polymer, which is meaningful to develop promising radioprotector in a more efficient way.

Answer: Thanks for the insightful question. In this research, we also hoped to summarize some rules to select functional groups to enhance the antioxidant capability of Hantzsch ester. Thus, we used aldehydes with well-known antioxidant substituents (ferrocene, nitrobenzene, and phenol groups) and benzaldehyde derivatives containing electron-pull and electron-draw groups to prepare monomers and polymers. Phenol and ferrocene groups are efficient in improving antioxidant ability of polymers. However, the antioxidant ability of polymers is related but not equal to their anti-radiation capability. Cellular experiments are necessary to select radioprotective polymers from antioxidant polymers, and no law seems to be found by the current data. We believe a larger polymer library combined with theoretical calculation will be helpful to find some rules to guide the development of polymer anti-radiators in a more efficient way. We will try it in our future research. Some of this discussion has been added in revised 'Challenges and possible extensions'. Thanks a lot for your illuminating questions!

Author's Response to Reviewer #2:

1) Please revise the writing style of the word "ionizing" must be the same all over the manuscript.

Answer: Thanks for the suggestion. The word 'ionizing' has been used in the revised manuscript as required. Thanks!

2) The high light paragraphs in the results and discussion should be preferred to remove to the experimental section.

Answer: Thanks for the suggestion. The high light paragraphs have been removed to the supplementary information as suggested. The manuscript looks more concise and better, many thanks!

3) More discussion and references are needed concerning with the level of cell necrosis, DNA damage, MAD, SOD and zebrafish larvae.

Answer: Thanks for the suggestion. More discussion and references about the level of cell necrosis, DNA damage, MDA, SOD and zebrafish embryos and larvae have been added in the revised manuscript as required. Thanks!

Author's Response to Reviewer #3:

1. Which part of polymer is able to scavenge free radicals; mechanisms are unclear?

Answer: Thanks for the question. The 1,4-DHP moieties in polymers should be the radical scavengers, which has been identified by the comparison of P(PEGMA) that only contains PEG segments and has nearly no radioprotection to cells. Moreover, radiation-induced ROS in cells under different culture conditions have been tested to study the possible mechanism of cell-protection. Both amifostine and P(2)(1) scavenged radiation-induced ROS in cells, resulting in partial and nearly complete cell-protection from radiation damage, respectively. This information has been added in the revised manuscript (Fig. S14). Thanks!

2. Colony assay is necessary to do for evaluation of cellular radioprotective effects.

Answer: Thanks for the professional suggestion. Colony formation assay has been carried out as required. The results agreed well with those obtained by other methods, they have been added in the revised manuscript (Fig. S10). Thanks!

3. Amifostine is a prodrug (thiophosphate part of molecule should be hydrolyzed *in vivo* to active form) then amifostine is weak *in vitro* for radioprotective effect, its free thiol derivative should be used.

Answer: Thanks for the professional suggestion. WR-1065 (the free thiol derivative and active form of amifostine *in vivo*) has been tested for cell protection as suggested. Compared with amifostine, WR-1065 showed similar and slightly weaker protection to cells under 80 Gy X-ray condition. A possible explanation is the high dose of radiation used in the current research, which might accelerate the oxidation of free thiol in WR-1065, leading to quick failure of its radioprotective ability. The experiment result of WR-1065 has been added in the revised manuscript (Fig. S13). Thanks!

4. Survival of irradiated mice (with and without treatment) should be used as standard *in vivo* model for demonstrating of radioprotective effect and calculation of DRF for both amifostine and polymers.

Answer: Thanks for the constructive suggestion. Zebrafish has been used as an animal model to study radioprotectors including amifostine. Zebrafish has an advantage in quickly and easily detecting many indicators related to radiation-induced injuries (death, deformation, abnormal organs and damaged DNA), some injuries (deformation, abnormal organs and damaged DNA) are difficult to be studied by using mice. Thus,

we chose zebrafish in this proof-of-concept research. Related literatures using zebrafish in radioprotection research have been added in the revised manuscript (Ref. 48-53).

We agree with you that mammals are better than zebrafish as animal models for pharmaceutical research because they are closer to human and more suitable to simulate clinical treatments. In our future research, we will try to develop polymers with enhanced radioprotective ability and biocompatibility. Mice and rats will be used to study not only anti-radiation capability but also pharmaceutical parameters (e.g. administration route/frequency, pharmacokinetics, pharmacodynamics and drug distribution) of these polymers. This discussion has been added in revised 'Challenges and possible extensions' section. Many thanks!

DRF is a great suggestion. We studied the relationships between different doses of X-ray and cell viability, hatched zebrafish larvae number and deformed zebrafish larvae number (Fig. S11, Fig. S17). These data were used to calculate the DRF values of amifostine and polymers at their cytosafe concentrations in cellular and zebrafish experiments. These results have been added in the revised manuscript. Many thanks!

We declare no competing financial interests in this research. The data supporting the findings of this study are available within the paper and its supplementary information files. The source data underlying Figs 2, 3c-5, 4b, 4c, 5b and Supplementary Figs S6, S7, S8, S9, S10, S11a, S11b, S13, S14, S17 are provided as a Source Data file.

REVIEWERS' COMMENTS

Reviewer #1 (Remarks to the Author):

The authors have revised the manuscript according to my comments, I think it can be published in Nature Communications.

Reviewer #3 (Remarks to the Author):

1-Colony assay is needed at different doses of radiation for survival fraction curve.

2-Survival of irradiated mice (with and without treatment) should be used as standard in vivo model for demonstrating of radioprotective effect and calculation of DRF for both amifostine and polymers.